# Designing self-tracking experiences: A qualitative study of the perceptions of barriers and facilitators to adopting digital health technology for automatic urine analysis at home

**Margherita Motta[1], Emily Groves[1], Andrea Schneider[1], Samantha Paoletti[2], Nicolas Henchoz[1], Delphine Ribes Lemay[1]***

**1** EPFL+ECAL Lab, EPFL, Lausanne, Switzerland, **2** Life Science Technologies, CSEM SA, Landquart, Switzerland

* delphine.ribes@epfl.ch

## Abstract

Self-tracking technologies open new doors to previously unimaginable scenarios. The diagnosis of diseases years in advance, or supporting the health of astronauts on missions to Mars are just some of many example applications. During the COVID-19 pandemic, a wide range of self-monitoring protocols emerged, revealing opportunities but also challenges including difficulties in understanding how to self-use monitoring systems, struggling to recognize the benefit of such systems and a high likelihood of abandonment. In this paper, we explore the role that design plays in the creation of a user experience of self-tracking, with a focus on urine analysis at home. We investigate adoption factors and forms of data expression to overcome the presented challenges. By combining insights from related work, semi-structured interviews and indicative user-tests, we show the potential of pairing a traditional numerical data representation (data quantification) with a qualitative expression of the data (data qualification). Indeed, qualitative expressions have the potential to convey the complexity of the phenomena tracked, enabling deep meaning-making and emotional connection to personal data. At the same time, we also identify issues with this approach, which can require a longer learning curve and lead to rejection by users more accustomed to traditional, numerical approaches. Based on the results, several recommendations have been converted into an experimental proposition, which also presents future plans for the continuation of the project. This article presents the first fundamental step in creating a meaningful experience of self-tracking, taking into consideration the needs and expectations of future users.

## Author summary

In this study, we focused on urine analysis at home and explored the role of design in shaping the user experience of self-tracking. By examining adoption factors and data

**Data Availability Statement:** Data used can be accessed here: https://doi.org/10.5281/zenodo.8113655.

**Funding:** This research was funded by the Swiss Innovation Agency Innosuisse (grant 55059.1 IP-ENG). NH was the recipient of the Innosuisse grant. Innosuisse provided support in the form of salary for MM, EG, AS and DR. SP and NH did not receive any financial compensation for their contribution to this research. The funders had no role in study design, data collection and analysis, decision to publish, or preparation of the manuscript.

**Competing interests:** The authors have declared that no competing interests exist.

expression, we aimed to address the complex challenges associated with these technologies. Our research drew from existing literature, conducted interviews, and performed user-tests to uncover the potential of combining numerical data representation with qualitative expressions. Our findings contribute to understanding the role of design and user experience in urine tracking technologies. We addressed emotional and functional aspects of different design approaches, highlighting the importance of guidance, discretion, personal identification, and engagement in urine home monitoring. In doing so, we addressed gaps in the research, which had previously underexplored the emotional impact of interface design and the user experience associated with urine tracking technologies. Our exploratory study lays the foundation for effective and user-centered urine tracking technologies and provides a valuable starting point for future research.

## Introduction

The recent rise of self-tracking technologies, also known as "Quantified Self", offers new opportunities in terms of personal agency, prevention medicine, and control. With the ever-increasing quantity of sensors that are placed in our phones, houses, and even clothing, the concept of self-tracking using digital technologies has become a common way to monitor, measure and evaluate the ways in which people conduct their lives [1]. This rapid popularity led to the emergence of a specific term, mHealth ('mobile health'), which was coined in 2003 [2] to describe the practice of using portable devices, as well as other emerging technologies, to monitor and measure personal health and well-being. As per November 2022, over 54'546 mHealth apps are available in the Google Play Store [3]. These applications offer the user a wide range of medical, health and wellbeing information from sleep patterns [4] to menstruation tracking. Central to these apps is the display of the information to the user, which can be achieved in various ways that frequently include the representation of the data into graphs and other forms of data visualization [5,6]. Self-tracking is one of the terms that refer to the practice of using data to structure and represent various aspects of human experience, but many others exist, such as personal informatics, personal analytics, lifelogging and the Quantified Self [7].

The topic of mHealth is vast and the possibilities are numerous. Motivated by the work of CSEM (Swiss Center for Electronics and Microtechnology, https://www.csem.ch/), a Research and Technology Organizations (RTO), on developing new home-based medical practices, in this paper we focus on the potential and challenges of using quantified self technologies for the tracking of biometric data, especially in the context of urine analysis at home. The increased demand for remote monitoring during the COVID-19 pandemic, which underscores the significance of this context, made us decide to further pursue the research with an user-centered approach. We worked with the scenario of a device plugged into the toilet at home, which can automatically measure biomarkers in urine such as pH, glucose and sodium, and which sends the results to a smartphone app. The selection of biomarkers was determined by their compatibility for integration into a toilet-attached portable device and their significance in relation to urine analysis. We are interested, here, in the design features and the associated experience facilitating or preventing the adoption of such a technology.

As the subject arouses many opinions, positive as well as negative, we describe here the initial design research steps to elucidate the drivers of adoption for mHealth technologies. In this process, grounded in empirical research, we followed the approach of research through design [8]. This includes investigation to gather diverse perspectives on the problem at hand, reflective practice and the integration of design activities and artifacts as a means of inquiry and

knowledge generation [9]. The paper starts by describing the related work and the opportunities and challenges of using Quantified Self technologies for the tracking of biometric data with a focus on urine analysis at home. This description is organized into five categories summarizing the most prominent issues linked with the Quantified Self phenomenon. Then, we present the results of a thematic analysis from data collected via semi-structured interviews performed with 16 people on the topic of urine self-analysis at home. From the related work and thematic analysis, we then propose potential design ideas to address the identified research gaps. Finally, we present the results of a user-tests done with low-fidelity prototypes on the proposed design ideas and, accordingly, we outline future potential design directions. The objectives are to help researchers and designers in their design journey to create a user experience for self-tracking technology in general and self-tracking of urine biomarkers at home in particular.

## Related work

### Challenges and opportunities of self-tracking technologies

We grouped the themes of the related work in five categories: intention, neutrality, embodiment, engagement, and privacy and security. By organizing it into categories, we aimed to provide a clear and concise summary of the most prominent issues in the field. Our goal was to gain a broad understanding of the state of the field and to identify potential areas of focus for future research.

"Intention" contains the motives that drive users to use self-tracking technologies. Some self-trackers decide to monitor data about themselves just to satisfy a personal curiosity or to remember aspects of their lives, while other people might collect personal information to set and achieve specific goals [10]. In the latter case, data is often used as a means to alter daily routines and habits that are deemed unhealthy or harmful, and so aiming at improving the quality of life of the user [11]. In the field of mHealth, the motivation for self-tracking often lies in the research for behavior change [12]: monitoring one's health becomes a means to improve wellbeing. In this sense, self-tracking practices both make people feel more in control of their lives and make their body feel healthier or stronger, while, at the same time, nurturing the feeling of awareness about themselves. As Choe et al. describe [13], such desires are frequently born from a concrete health-related need, such as executing a prescribed treatment or finding triggers for an allergy, rather than a general aim to become more healthy. At the same time, other studies show how, independently of the starting incentive, long-term self-trackers value the support and motivation that these technologies trigger for having a durable change in their lives [14].

"Neutrality" is the awareness that design choices are never neutral and shape the experience of the users. Self-tracking is based on quantification, often using advanced algorithms to process and display the data collected. However, the data itself does not inherently have any meaning until people examine it and assign significance based on their personal context and expectations [15]. The people who retrieve the data establish a relationship and a meaning between the information and the contexts in which they were generated according to who they are [16]. By creating artifacts like data visualizations and interfaces, designers can mediate the user's access to data and information [17].

"Embodiment" refers to the various relationships that self-trackers form with their personal data. As just discussed, self-tracking technologies can drive their users to form a new subjectivity and a new meaning of themselves and the world. Nevertheless, this relationship is ambivalent. At the same time, users also shape technologies by giving them a meaning by integrating them into their lives [10].

When regularly used, it is argued that self-tracking technologies amplify the capacities of the human body, as they supply the user with data that can be both saved and directly used to improve—or simply change—the body in the desired direction [5,13]. In this sense, users tend to form a relationship with the tracking devices that lead to complex forms of embodiment that shape how they control and manage their bodies. Over the last decade, this theme has become particularly prominent in the literature [5,7,13,18–21]. Lupton [15] argues that self-tracking technologies might detract users from self-agency, pushing the action and deliberation power from the user to the devices. Research conducted by Smith and Vonthethoff claims that the figure of the body and the self that emerges from constant practices of self-tracking and monitoring personal data alters the relationship between the user and its personal intuitions, meaning the ability of listening and understanding personal sensations [11]. In these frameworks, information that is considered reliable and trustworthy regarding their own body becomes outsourced. An example can be found in the study conducted by Homewood et al. [22], that used technological removal as a method to evaluate the impact of personal devices on users' lives and to facilitate the lived experience of self-tracking from an emotional, embodied, and cultural perspective. Findings of the study show how removing menstrual cycle tracking apps from the user's lives changed the experience of their bodies, initially losing certainty, but eventually making the subjects aware of their capacity to read their personal sensations. Other research, instead, gives self-tracking devices an empowering capacity, giving people personal agency and making them feel in control of their lives [7]. Feedback loops [14] have the potential to not only motivate users, but also to enable them to achieve results they may have previously believed unattainable. By consulting their own personal information, and by unconsciously accepting the responsibility that comes with it, self-trackers can exert control on their own bodies and lives. Through detailed monitoring and measurement of their lives, these visions tend to position such devices as so powerful to be able to improve human lives [19].

"Engagement" describes the relationship between a user's curiosity in self-tracking technologies and the tendency to abandon them. One of the most important open questions regarding tracking devices is how to engage people over a long period of time [6]. Abandonment is still a prominent issue in the user acceptance of Quantified Self devices [23,24]. Shin and Biocca [6] claim that one of the main problems with Quantified Self devices can be found in poor user experience. Although usability issues have not yet been fully addressed by the literature, the authors argue how they could be linked to poor user acceptance. This topic is strongly linked to the quality of the content: people's expectations for meaningful content are always increasing, and it's ever more necessary to design compelling experiences. As studies about affordance, the discipline that investigates subject agency and technological efficacy, also remind us, we must always consider that people—and their behaviors—change and evolve over time [13,25]. In different moments or stages of their lives, users might have different intentions, feelings, interests or concerns regarding themselves and their wellbeing [23]. The way in which affordance is structured is essential in defining the level of users' engagement, especially when attention is paid to what kind of information is collected and how it is presented to the user [25]. Results show that perception of quality and the presence of comparative feedback are linked to more prominent engagement and motivation in the users. In the health and wellness fields, in particular, users' pre-existing health consciousness was found to play a major role in the solidification of new health-related behaviors and consequently the tendency of staying engaged and continuing to use the product [6]. The study conducted by Epstein et al. on users that abandoned tracking technologies, show how some of them feel a sense of guilt and wish they had found the will or time to start the practice again [23]. These findings suggest that designers should explore ways to facilitate reengagement and research how their work contributes to a more successful experience.

"Privacy and security" regards the protecting of the user's personal data and the perception of the dangers of self-tracking practices. The practice of collecting data has gained a commercial and governmental value for many actors [26], and some work is questioning the way in which this data can be exploited [10,16,27–30]. The practices associated with capitalizing on the opportunities brought by collecting personal data, pushed by a new logic of accumulation and bypassing laws and social norms associated with privacy, have been termed "surveillance capitalism" [31]. Small and big companies tend to obscure their operations, hiding how personal data is stored and transmitted and, therefore, the way in which this information can be accessed by third parties either legally, through buying and selling, or illicitly through, for example, hacking [7]. Hepp, Alpen and Simon [32] analyze the way in which public discourse about the Quantified Self movement differs significantly from its internal self-perception. While the media frequently depicts the act of self-tracking as a problematic desire for personal control, or an open door for higher levels of dystopian surveillance, the movement tends to see self-measurement as an act of personal empowerment [29]. By pushing this idea of personal agency, tech companies promote the idea of tracking tools as choosing subjects whose well-being depends on and derives from the market choices they make. The risk in this process is the fact that consumers are invited to select technologies as if the world's stakes are in their hands, without having the skills or the knowledge to make accurate decisions. In the mHealth field, the problem becomes even more relevant. It is not just a matter of becoming a product for third parties, but the extent might go into a logic of subjugation, personal control and even a new kind of sovereign power [28,31]. As we discuss while talking about intention, neutrality and embodiment, self-tracking is strongly connected to a feeling of responsibility towards oneself, both in terms of wellness and self-identity. This responsibility can bring positive change into some subjects, but some authors [10,27,28,33] assert that such responsibility could be overturned into a tool of power in the hands of private companies. Particularly, when talking about health, this could give insurance companies the possibility of outsourcing costs and responsibilities of health prevention [10,31]. On the other hand, other authors define the act of self-tracking data as a form of soft resistance [11] to the monolithic health promotion rethorics endorsed by the establishment. This means a way to regain power over our own health. Although this paradigm might be idealized, the feeling of confidence, self-empowerment and overall well-being that most long-term self-trackers report experiencing should not be dismissed as unreal or insignificant. As users are usually unaware of the way their personal information is used, stored or even sold [29], design should provide transparency regarding the data usage of each technology. Design also plays a role in the creation of a trusting environment that, combined with transparency, allows users to safely get the most of mHealth technologies.

**Key concepts from related work.** The following paragraphs provide a summary of the five categories of challenges identified in the related work of self-tracking technologies for the tracking of biometric data, with a specific focus on user perceptions and experiences. These shall later be addressed in the design of the user experience for self-tracking technology.

- **Intention:** Different users choose to self-track for different reasons and to obtain different goals. These different intentions require different approaches to be best satisfied. Understanding the intention that future users might have is a fundamental step that needs to be taken into account.

- **Neutrality:** Data, even when composed of just numbers, is not a neutral item. In fact, it is one of the main things that defines the experience of the user in the context of self-tracking technologies. As researchers or designers, the choice of the data to track, to display and how it is shown tacitly decide what information and visualizations are insightful, what instead are not.

- **Embodiment:** Self-trackers create various complex relationships with their data. While various interpretations exist regarding this relationship, it can be argued that the exchange within this dynamic is often perceived as divisive, encompassing both empowering and disempowering aspects. On one hand, it can be seen as an empowering energy that enables the user, on the other hand as a limit for the user in their exchange with its body and the world.

- **Engagement:** Quantified self technologies easily attract the curiosity of users, but this often quickly wains, decreeing the failure of projects. When designing for devices that will be used over time, how the features are designed will prompt and deter the user interest and engagement in different phases. By analyzing user's behaviors and motivation, one can learn how to motivate people to track themselves and how to keep motivated during long periods of time.

- **Privacy and Security**: Data privacy and personal security are problems frequently addressed in the discussion around self-tracking. To be able to create a device that people might trust and put in their homes, we must ask ourselves how we can practically address this within the design. Concrete actions in developing systems that actively protect the user's data must be taken, but trust should also be built by explaining these choices in the communication system.

## Self-tracking technologies in the context of urine tracking

Several studies on the development of urine tracking devices have been presented in recent years [34–38]. However, little can be found regarding the perception that people have about the use of urine for medical practices, and the user experience concerning such technologies. This is further compounded by the fact that the aforementioned papers often present ideas or prototypes, without any information regarding the user's point of view. When presenting their mountable toilet system for the analysis of excreta, Park et. al. discuss briefly the taboo around human waste, assuming difficulties in user acceptance and compliance in case of widespread use of their technology [36]. They conducted a survey where the majority of responses received showed users to be either 'somewhat comfortable' (37.33%) or 'very comfortable' (15.33%) to use the proposed tracking system. However, the same authors argue that the results might be biased as it was conducted in a highly educated group and purely hypothetical, i.e. without actual device testing. Omoronyia et. al. conducted research on the comparison of urine and blood testing for HIV [39] as blood has always been the standard for such testing, but urine allows for a less invasive and risky procedure but with equal precision. Despite this, the study didn't find any statistically significant preference in the participants and a large proportion of the subjects were found indifferent to which body fluid was used for the testing. The authors, however, suggest that such results could be linked to a lack of education and awareness regarding the potentials of urine testing. An interesting contribution about the role of self-tracking of waste is brought by Søndergaard, who discusses the matter in relation to menstrual blood. The author discusses how attempts to track body waste is not a way to fight the taboos and create new awareness, but instead a measure to further "civilizing and disciplining ourselves" and actually strengthening such taboos. In this sense, self-tracking of human waste becomes for the author a commercialisation of intimacy that remains stuck in the social terms of pure-impure, without opening our gaze to the complexity of the phenomena dealt with [40].

## Literature gaps and research questions

Despite the acknowledged importance of designers in creating effective self-tracking technologies, there remains a significant research gap in terms of specifying the necessary design

actions for achieving such outcomes. We identified two main research gaps. Firstly, to the best of our knowledge, no studies were found regarding the emotional and the functional performances of different approaches of graphic, interface or information design in the quantified self field. Secondly, on a more specific level, no studies have examined the role of design and user experience in the context of urine tracking technologies to evaluate potential barriers or factors affecting acceptance.

Based on these gaps, we identified two research questions:

1. What design approaches influence user experience and adoption of technologies for urine home monitoring?

2. What form of data representation is best adapted to the context of urine home monitoring?

## Materials and methods

Ethical approval was granted by the Human Research Ethics Committee (HREC 069–2022) of EPFL, Switzerland. An Information sheet was presented to each participant and written consent was obtained before each interview and before user-testing.

### Semi-structured interviews

The purpose of this qualitative study was to collect user opinions on the discussed urine-monitoring technology and serves as a foundation for initial design ideas. The study was conducted using semi-structured interviews. This method was chosen as it offers a balance between the flexibility of an open-ended survey and the straightforward focus of a structured, closed survey. The research aimed to investigate the following themes: users' relationship with self-tracking and the importance that those same potential users give to their health and the actions they undertake in this regard. To this end, 16 healthy participants were recruited who belonged to one of the following categories: potential target users (10 participants), project collaborators, both direct (actively working on the project, 4 participants) and indirect (employees of companies or associations collaborating with the project, 2 participants). The interviews were conducted using a script of predetermined but open-ended questions (see S1 Appendix—*Semi-structured interview script*). Off-script follow up questions were sometimes asked depending on the participant's response. All the interviews were conducted by a moderator, directly talking to the participant, and by a facilitator, taking notes. 13 interviews were conducted on Zoom, and 3 were face-to-face. They lasted between 30 and 45 minutes. The analysis was executed through a thematic analysis [41], that allows the orientation of the data collected and a categorization. Given that this study was conducted with a limited sample size of only 16 participants, we recognize that the findings may not be representative of the broader population. Nevertheless, the consistency and convergence of responses across participants suggest a significant level of coherence, providing meaningful insights into user attitudes and behaviors towards self-tracking and urine-monitoring technology. These findings have informed our team's subsequent research and design decisions.

### Indicative user-tests with low fidelity prototypes

From the related work and the semi-structured interviews, we designed, developed and implemented three low-fidelity prototypes. The purpose of the user-tests is to investigate two aspects of the relationship between users and their tracked data. The first aspect refers to cognitive understandability, which is the ability of users to comprehend different forms of data representation. The challenge of *neutrality* was a key consideration in the selection of this aspect, as we

aimed to examine how our design choices impacted the understandability of the results. The second aspect, meaning-making, was selected based on insights regarding the challenges users face in forming meaningful connections with their data, particularly those related to *intention*, *embodiment*, and *engagement*. Users often struggle to make sense of the data they track and are more likely to engage with their data when it is presented in a way that is meaningful to them. Therefore, we aimed to investigate the quality and potential of the relationship between users and their tracked data by exploring their capacity to relate to different forms of data and form a meaningful connection with it. By adopting an inductive approach we translated these aspects into three parameters that create the differences between the prototypes: levels of abstractions, complexity and numerical precision. Challenges associated with *privacy and security* were not addressed at this stage of the project due to the low fidelity nature of the prototypes. 12 participants were asked to look at their dummy tracking data and provide verbal narration of what they were seeing and how it made them feel, as well as fill in questionnaires and participate in a final interview to evaluate the experience. The three different prototypes were developed to explore participants' reactions and understanding of different levels of qualitative expressions of the same self-tracking experience. We decided to have three sets of results with different grades of good and bad values for each prototype. The design of the prototypes ranged from a more standard quantitative way of displaying data towards a more interpretative approach. The difference between the prototypes can be seen in Fig 1.

**Prototype 1—The chart (Fig 2).** As mentioned by Lockton et al. [42], the use of plain numbers to build common maps, graphs and diagrams has become a default mode for the representation of information and data. This approach is commonly referred to as quantification. Following this approach, the first prototype represents a standard self-tracking app that allows a quantified consultation of personal data. The prototype presents two types of interface to be consulted: the single page of a result (three different days are available for consultation), and the archive. Because of its familiar nature, this prototype is used both to evaluate the strength of a quantified approach and as a baseline to compare the other prototypes.

This prototype is associated with the use case in Fig 3, designed to mimic a situation where the user would compare a new tracking session to previous ones.

Furthermore, the prototype experience was introduced by the following scenario

*"One week ago, you received in the mail a device that you ordered to track urine, and you are still in the phase of exploring this device and learning how to use it. This morning, when you woke up, you decided to do a measuring. After a couple of hours, during a coffee break at*

| | Prototype 1<br>The Chart | Prototype 2<br>The Star | Prototype 3<br>The Dome |
|---|---|---|---|
| Type of technology | Digital | Digital | Analog |
| Data representation | Visual | Visual | Sound + visual |
| Data's reading complexity | Low | High | Very high |
| Data's qualitative expression | Low | Medium | High |

**Fig 1. Chart that overviews the four parameters that were used to differentiate the prototypes.**

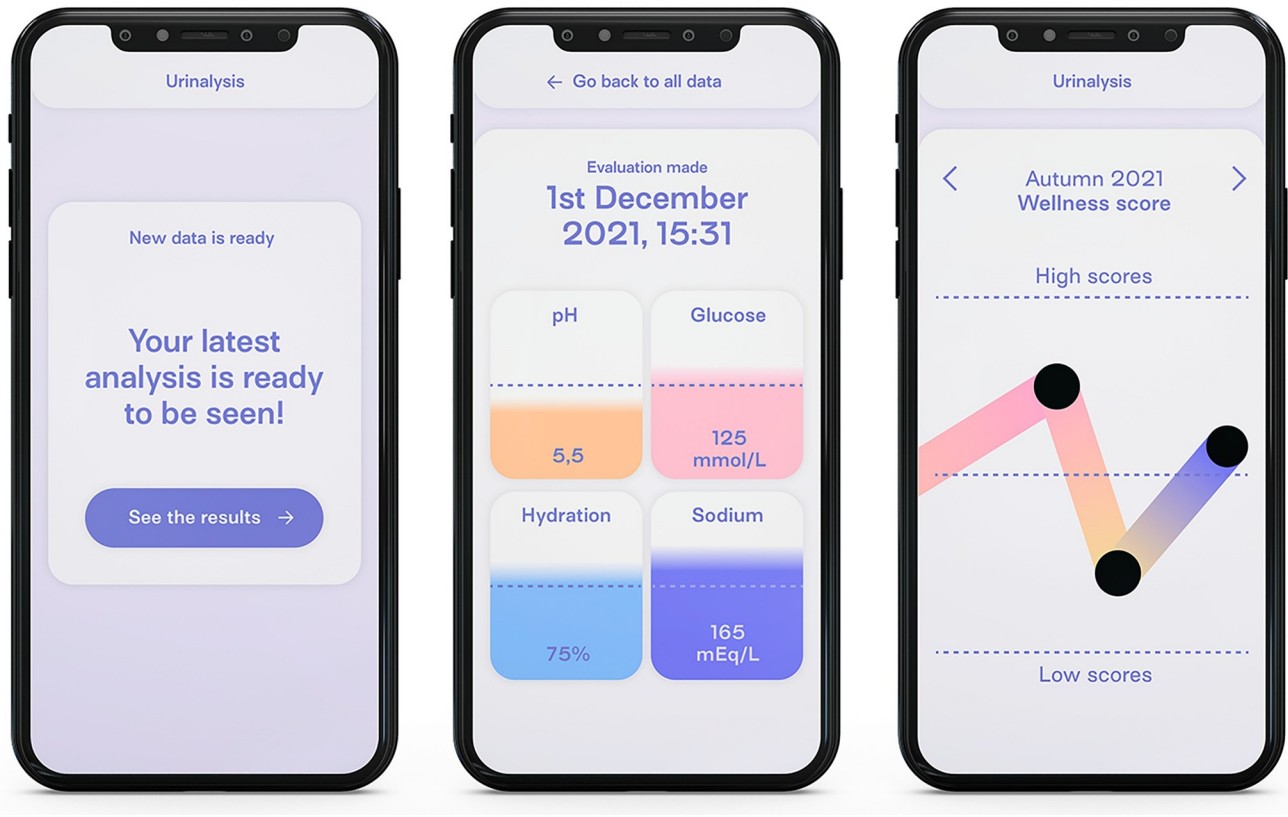

**Fig 2. Overview of the Prototype 1: In the Initial screen (1), the results page (2) and the archive page (3).**

*work, you decide to take a look at the results, so you open your phone to check it. After you have seen your results for the day, you decide that you want to go back and see other tests that you did in the last week and compare them.*"

**Prototype 2—The star (Fig 4).** The second prototype works as a link between quantitative and qualitative data representation. As the previous one, this prototype mimics an app for the consultation of tracked data, but a level of abstraction and complexity has been added. This prototype was developed to address the issue of self-trackers's relationship to their bodies and their selves, and the possibility of making links between information and personal body

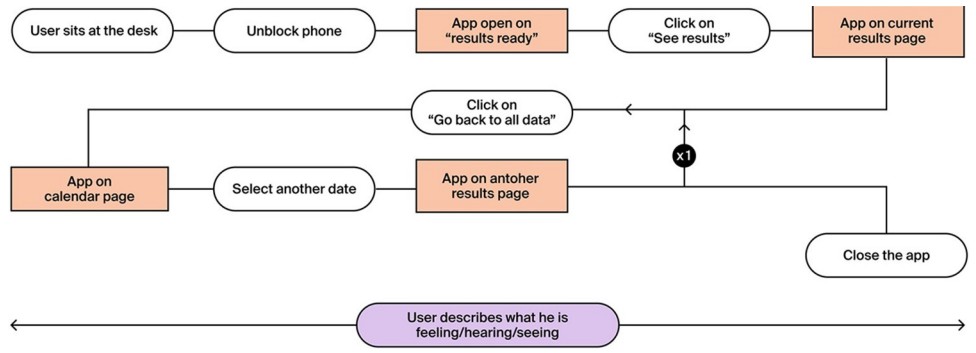

**Fig 3. The flow-chart associated with the Use case 1 and 2.**

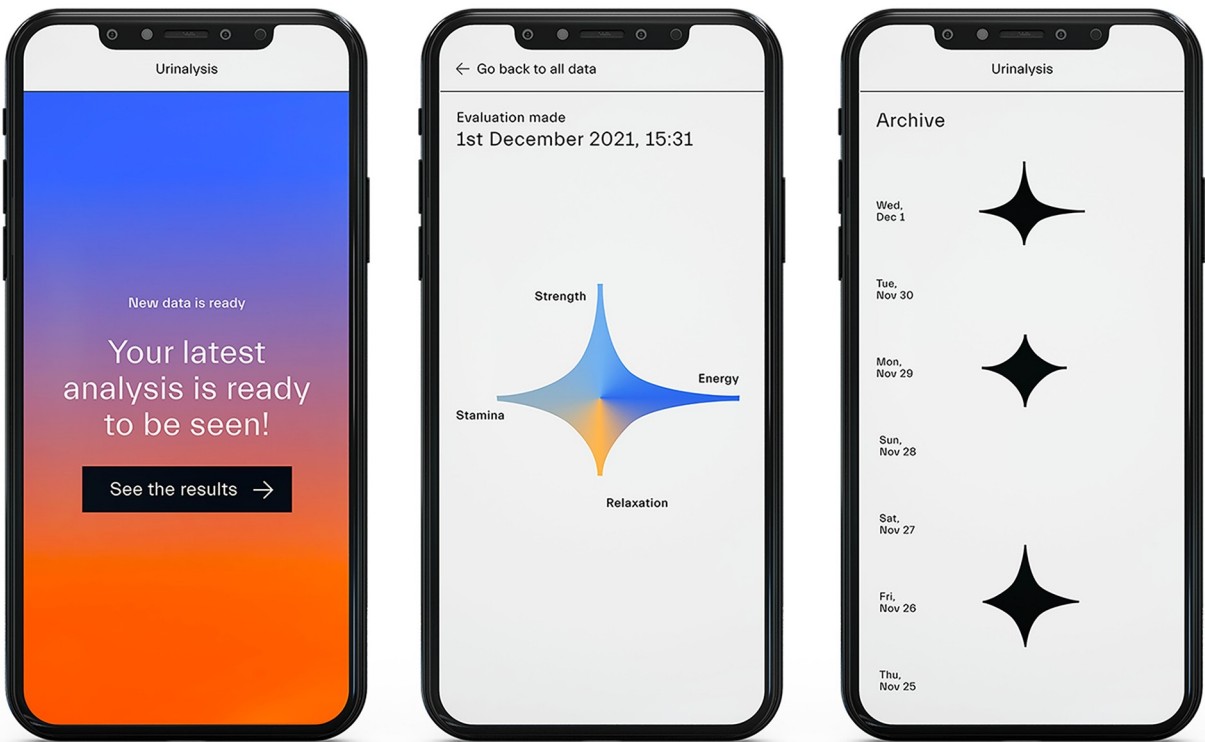

**Fig 4. Overview of the Prototype 2: In the initial screen (1), the results page (2), and the archive page (3).**

sensations [15,43]. The passage is represented by the replacement of the scientific name of the tracked parameters with names of emotions or feelings that are influenced by those same biomarkers. In this way, the prototype tries to push the user to make meaning out of the results they are presented and thus create a relation with their own daily life.

For the second prototype, both the use case and the scenarios are identical to the ones of the first. The reason is given by the fact that the difference between the prototypes lies, as previously discussed, in the level of abstraction of the graphical interface.

**Prototype 3—The dome (Fig 5, Fig 6).** The third prototype aims at exploring the concepts of data materialization and sonification paired with a qualitative interface. This experimentation wants to try to evaluate if such a design approach can balance frequently discussed problems with self-tracking technologies such as difficulty to understand the results [13] and loss of engagement [24]. In particular, this prototype was motivated by the possibility of turning data into physical artifacts which is discussed by Jensen et. al. [44]. Specifically, the prototype presents as a physical object with a cylindrical base and a semi-transparent spherical cap. The visitors are asked to wait for the activation of the device, and then carefully focus on the presentation of the data through a sequence of four white sounds and colored lights. Each pair of sound and light describes the positivity of a data hypothetically traced by the device. High amplitude of sound and warm light mean a positive result, while, on the other hand, low amplitude and cool light have a negative meaning. After the first sequence, the visitor is then asked to repeat the experience two more times, mimicking different measuring days.

For this prototype, the use case is designed to test a physical device that leaves the user in a passive position of a pure listener of his own data (see Fig 7).

For the scenario, parts are used to represent a fictitious lapse of time between different measurements.

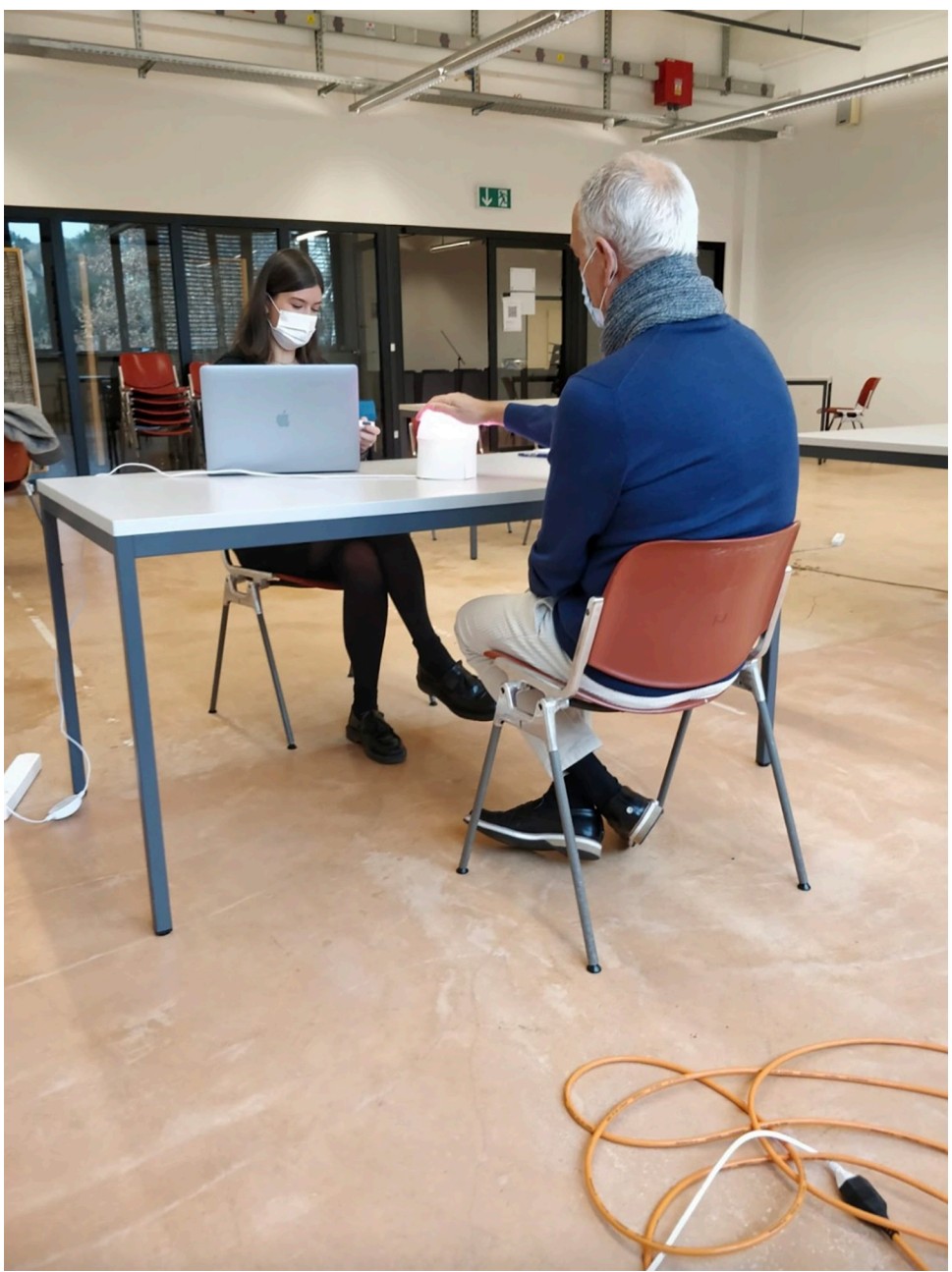

**Fig 5. Partial view of the testing of Prototype 3.**

Part 1: *"It is a weekend morning and you decide that it is time to take care of yourself. You decide to use your device, and while the results are being processed you decide to do other stuff around the house. Then you hear a sound and you understand that the analysis is ready, so you choose to take a look at all your results."*

Part 2: *"It's been a week, it is still a Sunday morning and you still want to keep track of your health and see how your results are compared to the previous week. As you've already done, you use the device and you wait to consult the results."*

Part 3: *"Again, it's been a week and it is time for a quick check. You again use your device and consult the results as soon as they are ready."*

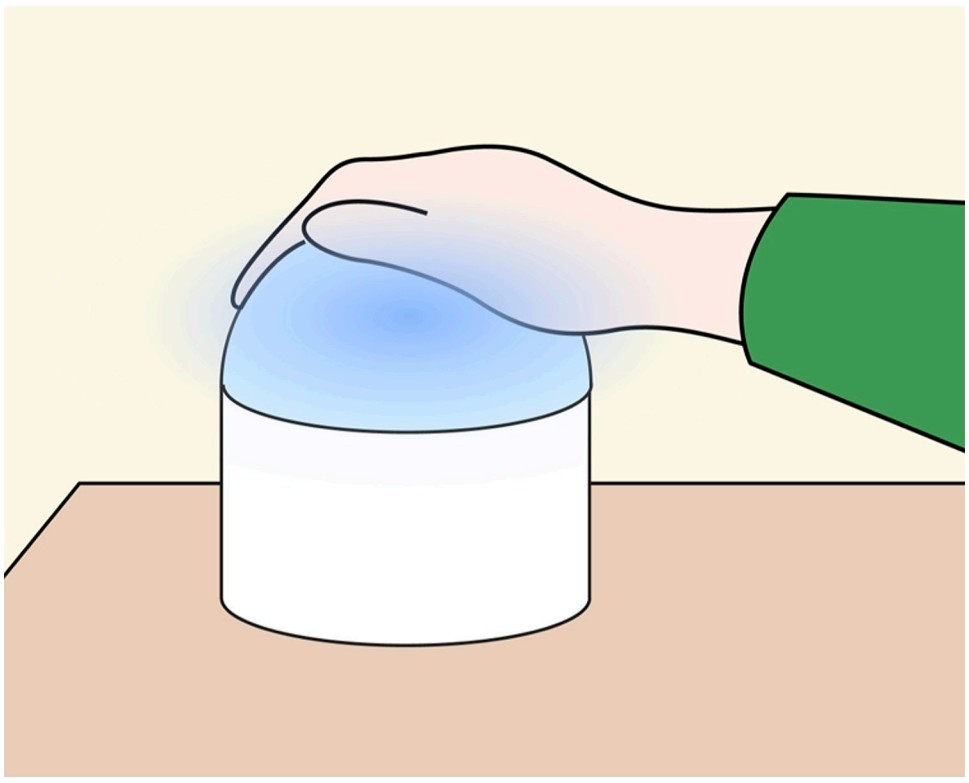

**Fig 6. Illustration of prototype 3.**

During the testing, both quantitative and qualitative data were collected. Quantitative data was collected through written questionnaires that were compiled by the user at the end of each test. The first part consisted of the Short User Experience Questionnaire [45]. Then participants were asked to answer questions on a scale from 1 to 7 related to data understanding, benefits from seeing past results and feelings and how they perceived data representation. Qualitative data, on the other hand, was assessed during a final interview and from the analysis of the verbal descriptions that the participants were asked to make during the whole experience. Questionnaires for each prototype can be found in the S2 Appendix.

## Results

### Thematic analysis from semi-structured interviews

The analysis of the interviews was executed through a thematic analysis [41] and using an inductive approach, which allowed for the themes to emerge organically from the qualitative data without imposing preconceived categories or theories. The processes produced several themes (see S3 Appendix), and five were chosen as the most compelling and conceptually

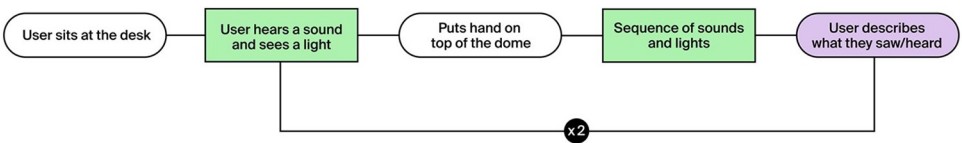

**Fig 7. The flow-chart associated with the Use case 3.**

enriching (need for guidance, motivation, data sharing, privacy and toilet sharing and effort and manipulation). These five themes were selected based on their ability to provide the most interesting and stimulating insights and are discussed in the subsequent paragraph.

**Need for guidance.** A common feature that interviewed self-trackers openly appreciate is having recommendations relative to the collected data. When using a tracking system, users desire feedback on their activity and suggestions on how to act to improve or maintain results. When given only the raw data, they might feel lost or bored by the tracking activity, while recommendations are valued as engaging and motivating factors. A user confessed to have stopped using a device because of this reason, and claimed to have asked himself "Here's my result, should I do something about it?". Although advice is highly valued, pushy reminders, especially in the form of notifications, can cause frustration and annoyance to the user. These feelings appeared to be present especially in relation to negative information or to already known and obvious personal sensations. For example, 2 participants mentioned feeling more stressed when their smartwatch notified them about increased heart rate and tried to push them to perform breathing exercises, causing the opposite of the desired effect. In this regard, the interviews also highlight a need for the user to be able to ignore recommendations and choose their actions autonomously.

**Motivation.** Half of the participants (8/16) mentioned motivation to perform as the main reason why they self-track. Some of them (4/16) said that tracking themselves and their activities is the only efficient way they found to keep positive habits. For sports, in particular, the ability to see concrete results while exercising, seems to create and makes the user proud of themselves. But not only does self-tracking help to keep habits, it also pushes the users to do more and achieve new and better results, creating a positive loop of self-affirmation and encouragement.

**Data sharing.** Around a third of the interviewed participants showed little interest towards sharing tracked data (5/16). The common reason seemed to be a lack of interest, although concerns about privacy were brought up by one user. Only one participant was highly positive about sharing. His feeling towards sharing is condensed in this quote "Look at me, I'm good!", that shows how sharing validates his effort. The same interviewed user mentioned having a group dedicated to data-sharing with his water polo teammates, used as a form of friendly competition.

**Privacy and toilet sharing.** Several subjects (6/16) presented immediate concerns regarding the possibility of other people seeing the device installed in their toilet. When asked to articulate, some users expressed a feeling of nuisance regarding the idea of explaining to potential guests what the device is used for. One user claimed that "Everything technological in the bathroom looks scary", while another stated, "I don't want my guests to see it and freak out!". Other subjects asserted that they share their toilet daily with members of their family and that they would probably not want something technological in the toilets. In addition, three subjects also expressed doubts on how the device would recognize the person that is using it each time. These doubts seemed to be connected to privacy issues. However, even if the majority of the interviewed were aware of the possible exploitation of personal data, the issue was not considered a deal-breaker in the use of tracking devices for themselves. At the same time, this awareness seemed to be projected onto possible guests or other people, leading to respondents' fear of making such people uncomfortable or scared while using a toilet that performs urine measurements.

**Effort and manipulation.** The topic of effort came out frequently during the interviews. For some participants (3/16), the effort necessary to track is sometimes unbalanced compared to the interest or relevance of the data received. In these cases, the loss of interest leads to the abandonment of the tracking activity. Related to this, a user said "I used a calorie tracking app

for two weeks, but it was too time consuming, complicated, and lacked joy". Another kind of effort was mentioned in regards to physical maintenance of the device. When they were described how the automatic urine analysis would work, a few participants (2/16) claimed that they would lose interest if the required manipulation and cleaning were too burdensome. While the concept of "too much" can be hard to define, it was implied that the problem would arise if the device's contribution to the quality of life would be sufficient to compensate for its monetary cost. In this regard, a participant stated that they would find the device uninteresting if it would require frequent manipulation to function properly. For the same participant, if the toilet device presented a low level of automation, then it would be "just like a Clearblue" (a famous brand of pregnancy and ovulation tests), making it pointless to have a complex and expensive device installed in your toilet. A good balance between costs, personal efforts and performances seems therefore to be essential to the self-tracking experience.

### Results of indicative user-tests with low fidelity prototypes

Twelve healthy participants aged between 23 and 70 years old (2 in the range 18–34, 9 in the range 35–54, 1 in the 55+ group) took part in the testing. The mean age was 43.8 years. Each participant tested three different prototypes (prototypes are described in the *"Materials and Methods"* part) that presented data in different ways and in a different random order. Every session was supervised by a moderator, talking directly to the participant and by a facilitator assistant, who discreetly took notes. The experience lasted between 20 and 30 minutes and always followed the same protocol.

**Quality of the experience.** The quality of the experience as perceived by users, divided by items of the Short User Experience Questionnaires (UEQ) [46], can be seen in Fig 8. The prototype 1 was rated as the most efficient (M = 5.5, SD = 1.09), clear (M = 5, SD = 0.8), conventional (M = 4.4, SD = 1.24) and usual (M = 4.3, SD = 1.43). On the contrary, the prototype 3

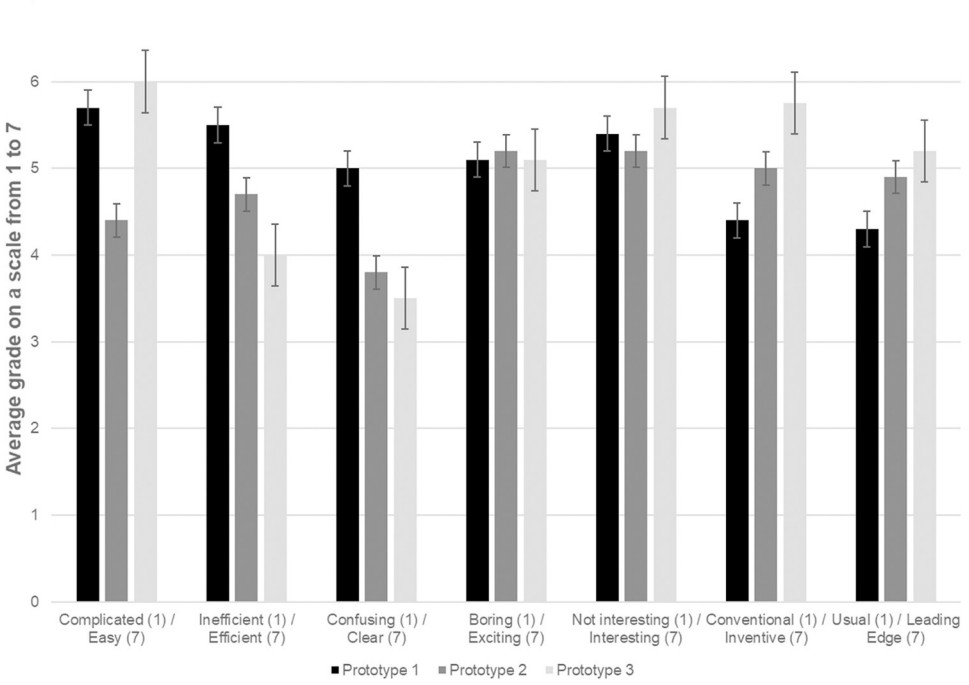

**Fig 8. Graph summarizing the UEQ written items for all three prototypes. Results are presented by their average.**

was rated as the most easy (M = 6, SD = 1.51), confusing (M = 3.5, SD = 1.79), inefficient (M = 4, SD = 2), interesting (M = 5.7, SD = 1.79), inventive (M = 5.75, SD = 1.61) and leading edge (M = 5.75, SD = 1.61). The prototype 2 is always rated in between prototype 1 and 3 for the items described previously.

**Data understandability and comparison.**   This section regards the understanding that the participants had of the data shown to them. On average, to the question "I understood the meaning of the results that I saw", where participants could respond using a scale from 1 (totally disagree) to 7 (totally agree), prototype 1 scored the best (M = 4, SD = 1.95), followed by prototype 2 (M = 3, SD = 1,09) and prototype 3 (M = 3, SD = 1.68). On average, to the question "Seeing past results made me better understand new ones", where participants could respond using a scale from 1 (totally disagree) to 7 (totally agree), prototype 3 (M = 6, SD = 1.81) scored the best, followed by prototype 1 (M = 5.5, SD = 2.27) and prototype 2 (M = 3, SD = 1.09).

**Multi-sensoriality.**   When asked if the sound sequences of the third prototype helped them to understand the data, users gave mixed responses. One third (4/12), gave the answer the best score, while another third gave either a 1 or a 2 (lowest score). When asked if "The lights helped me to understand the results" from a scale from 1 (totally disagree) to 7 (totally agree), all of the users gave a score superior to 4, the vast majority rated above 5 (11/12).

**Preferences and dislikes.**   During the interview conducted at the end of every testing session, participants were asked to articulate more their experience and express what they enjoyed and disliked regarding the prototypes. From these discussions, trends were extracted for each prototype.

For the first prototype, overall, all users expressed that the interface was the most familiar to them. Participants described this prototype as specific and efficient. 4 users defined the numerical results as "valuable" and as the most adequate way to display health-related results. At the same time, 2 participants explained how, even though the data visualization was familiar to them, they felt the lack of explanation of how the biomarkers relate to them and how those results affect their life. In particular, one stated "pH means nothing to me", expressing a lack of tools for interpretation of the plain results and the need for a more personal explanation of the data and parameters.

The second prototype received mixed feelings from the participants. The most common criticism refers to the difficulty of comparing data between different days expressed by 5 participants. Both the colors used and the shape were considered insufficient in explaining the results. Furthermore, the differences in the shape of the "star" and the connections of the axis were considered too subtle and not understandable. At the same time, several participants (5/12) appreciated the fact that the visualization expressed feelings instead of plain numbers. It "gives sense to the context" and maps personal states that the user can actually relate to. One participant preferred this prototype over the others as it is the "most interesting because it says how I feel".

In general, the third prototype was appreciated for its distinctiveness, uniqueness and fun atmosphere. Several participants (9/12) stated that they were excited from trying a different experience that was described by one as "almost a mysterious game" and by another as "more direct and personal". Two participants claimed that they preferred this prototype over the others as they already have enough apps on their phones, and they would appreciate having something different to consult personal data. At the same time, one third of the participants claimed that such a device could not exist on its own, but the experience should be paired with the possibility of going more in detail into the results. Furthermore, 2 participants expressed how they thought that the device would require a learning curve to be fully understood. 2 others also expressed criticism regarding the gamification aspect of the experience, claiming that such a choice could be inappropriate for health-related analysis.

## Discussion

Our study contributes to the existing body of research examining the influence of design on the impact of self-tracking technologies on users [6,13,23,26], but it also goes beyond by providing an analysis of different design approaches in the context of urine home monitoring.

In particular, the following gaps were identified and addressed in this study: (1) **Emotional and functional performances of different design approaches:** In today's world, the spread of self-tracking technologies is everyday wider and the competition becomes stronger. Therefore, it is important to understand the role that design plays in the adoption factors. Although some literature discusses the emotional perception of data representation [47,48], little is told about how design practices affect the mHealth's challenges previously discussed and, to the best of our knowledge, no studies that connected interface design with the user's emotional response were found. (2) **The role of design and user experience in the context of urine tracking technologies:** Despite numerous studies focusing on technical advancements and developments in urine tracking devices [34–38], there is a lack of research addressing the user experience associated with these technologies, resulting in a largely unexplored topic regarding the role of design in the context of urine tracking.

We addressed these gaps and gave answers to our research questions by combining the insights derived for the analysis done in the related work section with the ones derived from semi-structured interviews and low fidelity user tests.

Regarding our first research question: *What design approaches influence user experience and adoption of technologies for urine home monitoring*?, the following reflections were drawn and we suggest should be taken into consideration when designing mHealth technologies:

**Health is a topic that influences all age groups**: The majority of the participants of the semi-structured interviews (13/16) claimed to take some form of action in order to take care of their health, this is corroborated in literature [12]. **Guidance is an essential point of the tracking experience**: Users feel the need to be guided in both the understanding of the data and the action to take to improve their results. **Discretion is highly valued**: Users look at self-tracking like a private activity, which concerns and interests only themselves or some person very close to them. Understandably, this is even more important when talking about sensitive data. Discretion also implies that users are not eager to share personal data and that many need the assurance that their privacy is protected. **Personal identification enriches the experience:** It is important to design for personal identification, meaning that there should be options to personalize the experience. Doing so in terms of goals seems like a good path to follow. Given that our focus is to design a home device, this also implies that it should not be promoted for the monitoring of a third party. **Engagement is an open challenge:** Not only is it one of the most covered topics in the literature regarding self-tracking, but abandonment due to the lack of engagement was confirmed as an extremely common issue in all our research and has not been solved yet.

Regarding our second research question: *What form of data representation is best adapted to the context of urine home monitoring*?, we suggest one main finding that should guide further research.

**Seeing numbers is necessary, but not enough:** Existing literature has identified challenges that self-tracking users face in making sense of the data they collect [49] and our study further explores these difficulties in the context of health-related data. While our participants appreciated seeing their health data quantified, they also expressed a desire for the information to be presented in a way that was more personally relevant and understandable. Participants noted that medical terminology and raw numerical data can be difficult to process and apply in their daily lives, which may impede their ability to take meaningful action based on the information

presented. Therefore, we suggest that for urine home monitoring, data representation should balance numerical representation with qualitative ones, allowing for both precision and personal understandability of the information. Most current tracking devices use digital tools, such as apps, to present the data to the users. Typically, that data is represented using a straightforward visualization system made of numerical charts and tables. However, our work continues earlier efforts to demonstrate that there is more to the information that is being tracked than just their numerical representation [42]. As Morozov argues, numeric systems are ineffective in describing complex systems. A representation of the data that implies a form of narration, on the contrary, can allow the comprehension of such complexity [50]. In fact, phenomena that we choose to track are never one-dimensional, and, especially when they are related to the body, they imply a multitude of connections to actions and sensations experienced by the user. Extreme simplification, although apparently easy and satisfying, leads users to quickly lose interest in auto-tracking technologies [13]. Understanding the complexities and comprehensibility of phenomena can offer potential solutions to the common problems affecting technologies, specifically the ones highlighted in the related work section through the identification of five key challenges: *intention*, *neutrality*, *embodiment*, *engagement*, and *privacy and security*. Our approach towards this investigation was focused around the concepts of meaning-making [51] and cognitive understandability.

The results of our study suggest how, in the context of urine self-tracking technology, meaning-making and cognitive understandability can be achieved thanks to a qualitative expression of data paired with standard numerical-based information. Such a pairing can help to represent the complexity and the nature of the phenomena that is being tracked and favor a personal link between the user and the data. When done separately, as in the user-tests conducted here, users can get either frustrated or confused. By combining these approaches, we hypothesize that design could become the mechanism to both increase the quality of the user experience, generate positive emotions, improve engagement and limit problems such as technology abandonment and insufficient meaning-making.

Therefore, in line with previous research [47,48,51], we identified the human factor and the related emotional responses to the aesthetics, and user experience as a whole, as important components to consider in designing data interfaces. In our study, we expanded this idea to self-tracking technologies and the mHealth sector. We propose to include evaluation of user experience in the development of future mHealth applications to achieve acceptance of those technologies among the users.

As an exploratory study, our focus was on gathering diverse perspectives and generating preliminary insights to inform future investigations. While our insights are limited by the small sample for both the interviews and the user-test, it is important to note that the aim was not to draw definitive conclusions but to uncover emerging patterns and guide further research in the design process. Our study represents an initial step in the research process of designing for automatic urine analysis at home. It therefore serves as a foundation for subsequent iterations that would need to involve larger samples, higher-fidelity prototypes that closely resemble the intended final product, and extended observation periods.

## Conclusion

In conclusion, our study contributes to the existing body of research by addressing the gaps in understanding the role of design and user experience in urine tracking technologies through the evaluation of semi-structured interviews and a low-fidelity user test. We identified and addressed the emotional and functional aspects of different design approaches, highlighting the importance of guidance, discretion, personal identification, and engagement in the user

experience of urine home monitoring. Furthermore, we found that while numerical representation of health data is necessary in the context of urine home monitoring, it is not sufficient. Data representation should also incorporate qualitative elements to enhance personal relevance and understandability. We propose a balanced approach that combines qualitative expression with standard numerical-based information to represent the complexity of the tracked phenomena and establish a meaningful connection between users and their data.

While our study was not specifically aimed at producing generalized findings, it has established a valuable starting point for future research endeavors on the topic. Future studies will involve a larger and more diverse participant pool and employ high-fidelity prototypes for comprehensive evaluations. Additionally, conducting studies with prolonged exposure and usage will provide deeper insights into user perceptions over time. Overall, our study lays the foundation for further exploration and advancements in designing effective and user-centered urine tracking technologies.

## Supporting information

**S1 Appendix. Script used for conducting the semi-structured interviews.**
(DOCX)

**S2 Appendix. Complete questionnaire employed in the user test.**
(DOCX)

**S3 Appendix. Coding of the semi-structured interviews representing the identified themes, their details, examples and the number of occurrences.**
(DOCX)

## Acknowledgments

The authors thank Béatrice Durandard for advice on creating the design of prototype 3 and Cédric Duchêne for his supervision on the electronic parts. We also express gratitude to ECAL and Antoine Vauthey for providing us with the space to conduct the user test. The authors wish also to thank their project partners, ESTEE and CSEM, for the provision of the tracking device.

## Author Contributions

**Conceptualization:** Margherita Motta, Nicolas Henchoz.

**Formal analysis:** Margherita Motta, Andrea Schneider.

**Funding acquisition:** Nicolas Henchoz, Delphine Ribes Lemay.

**Investigation:** Margherita Motta, Emily Groves.

**Project administration:** Delphine Ribes Lemay.

**Resources:** Samantha Paoletti.

**Supervision:** Emily Groves, Delphine Ribes Lemay.

**Writing – original draft:** Margherita Motta.

**Writing – review & editing:** Emily Groves, Andrea Schneider, Delphine Ribes Lemay.

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
