## [Decision Letter · Decision Letter 0]

25 Apr 2023

PDIG-D-23-00007

Designing Self-Tracking Experiences: A qualitative study of the perceptions of barriers and facilitators to adopting digital health technology for automatic urine analysis at home

PLOS Digital Health

Dear Dr. Ribes Lemay,

Thank you for submitting your manuscript to PLOS Digital Health. After careful consideration, we feel that it has merit but does not fully meet PLOS Digital Health's publication criteria as it currently stands. Therefore, we invite you to submit a revised version of the manuscript that addresses the points raised during the review process.

Please submit your revised manuscript within 30 days May 25 2023 11:59PM. If you will need more time than this to complete your revisions, please reply to this message or contact the journal office at digitalhealth@plos.org. Please include the following items when submitting your revised manuscript:

We look forward to receiving your revised manuscript.

Kind regards,

Valentina Lichtner

Academic Editor

PLOS Digital Health

Journal Requirements:

3. Please send a completed 'Competing Interests' statement, including any COIs declared by your co-authors. If you have no competing interests to declare, please state "The authors have declared that no competing interests exist". Otherwise please declare all competing interests beginning with the statement "I have read the journal's policy and the authors of this manuscript have the following competing interests:"

4. Please provide separate figure files in .tif or .eps format only and remove any figures embedded in your manuscript file. Please also ensure that all files are under our size limit of 10MB.

5. Figure 3 and 5: Please confirm whether you drew the images / clip-art within the figure panels by hand. If you did not draw the images, please provide (a) a link to the source of the images or icons and their license / terms of use; or (b) written permission from the copyright holder to publish the images or icons under our CC-BY 4.0 license. Alternatively, you may replace the images with open source alternatives. See these open source resources you may use to replace images / clip-art:

- https://openclipart.org/"

6. Figure 5 includes an image of an identifiable person. Please provide written confirmation or release forms, signed by the subject(s) (or their parent/legally authorized guardian), giving permission to be photographed and to have their images published under our CC-BY 4.0 license. 

Otherwise, we kindly request that you remove the photograph.

Additional Editor Comments (if provided):

Please note that when Reviewer 3 refers to grounded theory, we take this to mean 'an inductive approach'. Grounded theory is a specific method and it would need to be appropriately referenced - we don't think you applied grounded theory.

Reviewers' comments:

Reviewer's Responses to Questions

**Comments to the Author**

1. Does this manuscript meet PLOS Digital Health’s publication criteria? Is the manuscript technically sound, and do the data support the conclusions? The manuscript must describe methodologically and ethically rigorous research with conclusions that are appropriately drawn based on the data presented.

Reviewer #1: Yes

Reviewer #2: Partly

Reviewer #3: Partly

2. Has the statistical analysis been performed appropriately and rigorously?

Reviewer #1: Yes

Reviewer #2: No

Reviewer #3: I don't know

3. Have the authors made all data underlying the findings in their manuscript fully available (please refer to the Data Availability Statement at the start of the manuscript PDF file)?

Reviewer #1: No

Reviewer #2: No

Reviewer #3: No

4. Is the manuscript presented in an intelligible fashion and written in standard English?

Reviewer #1: Yes

Reviewer #2: Yes

Reviewer #3: Yes

5. Review Comments to the Author

Reviewer #1: This is an excellent paper that discusses timely and significant issues related to self-tracking technologies with new empirical insights that are grounded on the emerging sociological literature on these issues. 

I think that the authors do a great job at situating their work in this literature, by presenting some of the most prominent themes that have emerged so far – this is potentially of great interest to the audience of this journal, that might not be aware of this work. 

On this basis, the authors have developed a mix of design work and interviews with participants self-tracking for urine analysis. This is very interesting and significant work, and my only suggestion here would be that I would like to know more on the ways in which results from the literature review actually guided the design of the three prototypes. 

The results that the authors collected from the interview and discuss here are also fascinating and very significant for the audience of this journal and beyond. The only suggestion I have on this point concerns the usefulness of raw data and numerical representations. I wonder if the authors can say more on what research participants felt was lacking from raw data and numerical representations and in particular whether they lamented a lack of epistemic tools to understand and process the data as actionable evidence (see e.g. a few papers on this in the context of wearable technologies: https://doi.org/10.1177/2053951720924436 ; https://doi.org/10.1371/journal.pdig.0000104).

Reviewer #2: This paper proposes best practice principles for designing an improved self-tracking user experience (UX) for general mobile health and for urine analysis tracking in particular. They highlight five important different aspects of the self-tracking UX and examine two approaches to measure these aspects. The first is a semi-structured interview to allow enough flexibility (I wished to read this interview but couldn’t find the relevant part in the text. Where is Appendix A?) and let 16 individuals answer it. The second is three different low-fidelity prototypes (two apps and one physical object) to investigate user understanding of their data and engagement level and try them with mock data on 12 individuals.

This paper gives a somewhat psychological point of view of what can increase users’ engagement and diligence. I have to say that I’m less proficient with such kinds of papers. My expertise is more computational and thus, my major comment would be regarding the number of interview participants. The authors did not mention how many actual potential target users took part in the interview part (as if 16 is not already extremely low). They only mentioned that they were part of the group (could be only one participant…). It could give some intuition about people’s preferences but drawing conclusions based on such a small dataset is neither sound nor robust IMHO. I believe the ms. could be improved if more interviewees were recruited for this study. That being said, the title of this ms. suggests that it’s not a quantitative survey but rather qualitative. Since the editor decided to continue with the review process I guess the ms. meets the journal’s quality bar and thus will recommend accepting for publication.

Few minor comments the authors may want to address are (I added line number when needed):

Figure 1 does not exist. 

Figure 6 is not very informative.

References are not consistently formatted.

46 As far as I know, unless it’s very seminal, a direct quotation is not a common convention in scientific papers. I would’ve removed it (given the correct reference) or re-write in my own words.

106 Check references numbering order (14->27..29->19)

145 user’s -> users’

182 “Privacy and security”regards.. Missing a space

484 redundant dot

574 mHeatlh typo

Reviewer #3: Thank you for the opportunity to review “Designing Self-Tracking Experiences: A qualitative study of the perceptions of barriers and facilitators to adopting digital health technology for automatic urine analysis at home.” I enjoyed reading this paper and compliment the authors on drawing together this important work. There is definitely more work to do on understanding the interface between the quantified-self, design, usability, data comprehension, meaning-making, emotional experiences and long-term engagement of users in health devices.

The paper is written very well and I have no comments about the writing except I noticed two typos which will be fixed in proofing. Also, I would reference the second sentence in the abstract as it reads like your opinion not objective data. 

I have a few general concerns about the paper which, in association with the other reviewers and the editor, you may wish to address. I am not sure this journal is actually the best place for this paper. This paper takes a very constructivist approach which may not suit a positivist context. However, I will let the editor make that final decision. None the less this is a high quality paper. Perhaps up front you can signal you are presenting a critical hermeneutic work that is grounded in empirical research. This is an unarticulated strength of this work. 

Related work

Firstly, from my knowledge of the literature in the quantified self space you have done an excellent job at representing the arguments which support and critique self-tracking, particularly in relation to power and control. The thematic synthesis of this review is sound. I personally would prefer to see the section “related work” (e.g. literature review) prior to seeing the context of the study (urine tracking). This would make for a cleaner reading of the review (theory) without going into the context (operationalisation of the theory). Then I would make a clearer argument as to why you chose the context. I believe it is in the paper, just pull it out and make it stronger. This would then provide a link between the literature and context that makes more sense and it wasn’t just a random choice.

Also in the end of the previous work section I would have expected to see the ‘gaps’ in the literature and some research questions you are trying to address.

Methods 

In the methods section I would like to understand more about the thematic analysis. It sounds like you used grounded theory, rather than an a-priori framework, however this should be articulated and codebook made available. 

It is not totally clear how you drew from related work and semi structured interviews (line 276) to develop the parameters of the prototypes. If you can make this clearer in a table which describes comments, codes /themes, design insights and prototype elements that would make for a better link between the literature, interviews and methods. 

Results

It is slightly confusing that the results of the semi-structured interviews are placed after the prototypes are introduced, given they informed the prototypes, however this may only be an issue for me. In the results section I might expect to see a return to the literature and address the ‘gaps’ in the literature and an answer to the research questions you proposed based on your literature search. Again I think the information is in the paper, but could be made more explicit. 

Conclusion

I think you have a done a good job at summarizing the work and as I work in a Centre which both researches and develop technology I can see how these would be useful for practitioners. 

Thank you

6. PLOS authors have the option to publish the peer review history of their article (what does this mean?). If published, this will include your full peer review and any attached files.

**Do you want your identity to be public for this peer review?** For information about this choice, including consent withdrawal, please see our Privacy Policy.

Reviewer #1: No

Reviewer #2: No

Reviewer #3: No

---

## [Editor Report · Decision Letter 1]

21 Jun 2023

PDIG-D-23-00007R1

Designing Self-Tracking Experiences: A qualitative study of the perceptions of barriers and facilitators to adopting digital health technology for automatic urine analysis at home

PLOS Digital Health

Dear Dr. Ribes Lemay,

Thank you for submitting your manuscript to PLOS Digital Health. We appreciate the changes made to the manuscript in response to reviewers' comments. However, we believe the manuscript needs further work before it can be published by this journal. 

We invite you to submit a revised version of the manuscript that addresses the points raised during the review process.

Please submit your revised manuscript within 30 days Jul 21 2023 11:59PM. If you will need more time than this to complete your revisions, please reply to this message or contact the journal office at digitalhealth@plos.org. Please include the following items when submitting your revised manuscript:

We look forward to receiving your revised manuscript.

Kind regards,

Valentina Lichtner

Academic Editor

PLOS Digital Health

Journal Requirements:

Additional Editor Comments (if provided):

The paper needs a further revision in the Discussion/Conclusion. The discussion must more clearly relate the literature to the current study; i.e. a discussion of this study in relation to what is already known. It may also discuss the limitations of this study. 

The conclusion should 'wrap up' and close the paper, possibly with key messages and further research. It should not add new findings. Consider moving the content of the conclusion to the discussion. 

A few minor edits also required across the other sections: 

Remove the citation from the Abstract [1]

Citations should start from [1] in the introduction

line25: traditionalist users - may be taken as judgemental; a more descriptive term could be used instead

55: CSEM - Microtechnolog - a typo? Please consider also providing a URL for more information on this Centre

69-70: 'we followed the approach of research through design' - needs a reference

104: 'self-trackers give a lot of value to..' - feels colloquial, consider changing to 'self-trackers value ... '

221: 'Although there are a lot of interpretations given to such relationship, we can affirm with certainty that the exchange present into this relationship is divisive, as it can be both seen as an empowering energy that enable the user' - the claim for certainty needs to be substantiated (consider rephrasing, and/or add references)

421: 'The majority of participants (8/16)' - this appears to be half, not a majority

533: 'Many papers note...' - needs references of these papers

561: '...technologies as seen in related work' - needs references to the related work
---

## [Editor Report · Decision Letter 2]

7 Jul 2023

Designing Self-Tracking Experiences: A qualitative study of the perceptions of barriers and facilitators to adopting digital health technology for automatic urine analysis at home

PDIG-D-23-00007R2

Dear Ribes Lemay,

We are pleased to inform you that your manuscript 'Designing Self-Tracking Experiences: A qualitative study of the perceptions of barriers and facilitators to adopting digital health technology for automatic urine analysis at home' has been provisionally accepted for publication in PLOS Digital Health.

Best regards,

Valentina Lichtner

Academic Editor

PLOS Digital Health